The EMBO Journal (2013) 32, 3119–3129
www.embojournal.org

# Cohesin-mediated interactions organize chromosomal domain architecture

Since Advance Online Publication, accession codes have been added and a change made to the Acknowledgements

**Sevil Sofueva**[1,6]**, Eitan Yaffe**[2,6]**, Wen-Ching Chan**[1]**, Dimitra Georgopoulou**[1]**, Matteo Vietri Rudan**[1]**, Hegias Mira-Bontenbal**[3,7]**, Steven M Pollard**[1,4]**, Gary P Schroth**[5]**, Amos Tanay**[2,]*** and Suzana Hadjur**[1,]***

[1]Research Department of Cancer Biology, Cancer Institute, University College London, London, UK, [2]Department of Computer Science and Applied Mathematics, Department of Biological Regulation, Weizmann Institute, Rehovot, Israel, [3]Lymphocyte Development Group, MRC Clinical Sciences Centre, Imperial College London, London, UK, [4]Samantha Dickson Brain Cancer Unit, Cancer Institute, University College London, London, UK and [5]Illumina, Inc., Hayward, CA, USA

To ensure proper gene regulation within constrained nuclear space, chromosomes facilitate access to transcribed regions, while compactly packaging all other information. Recent studies revealed that chromosomes are organized into megabase-scale domains that demarcate active and inactive genetic elements, suggesting that compartmentalization is important for genome function. Here, we show that very specific long-range interactions are anchored by cohesin/CTCF sites, but not cohesin-only or CTCF-only sites, to form a hierarchy of chromosomal loops. These loops demarcate topological domains and form intricate internal structures within them. Post-mitotic nuclei deficient for functional cohesin exhibit global architectural changes associated with loss of cohesin/CTCF contacts and relaxation of topological domains. Transcriptional analysis shows that this cohesin-dependent perturbation of domain organization leads to widespread gene deregulation of both cohesin-bound and non-bound genes. Our data thereby support a role for cohesin in the global organization of domain structure and suggest that domains function to stabilize the transcriptional programmes within them.

*The EMBO Journal* (2013) **32,** 3119–3129. doi:10.1038/emboj.2013.237; Published online 1 November 2013
*Subject Categories:* genome stability & dynamics; development
*Keywords:* chromosome organization; cohesin; transcription

*Corresponding authors. S Hadjur, Cancer Institute, University College London, Paul O'Gorman Building, 72 Huntley Street, London WC1E 6BT, UK. Tel.: +44 (0)2076790854; Fax: +44 (0)2076796817; E-mail: s.hadjur@ucl.ac.uk or A Tanay, Department of Computer Science and Applied Mathematics, Department of Biological Regulation, Weizmann Institute, Rehovot 76100, Israel. Tel.: +972 8 9343579; Fax: +972 8 9346023; E-mail: amos.tanay@weizmann.ac.il
[6]These authors contributed equally to this work.
[7]Present address: Department of Reproduction and Development, Erasmus MC, University Medical Center, Dr Molewaterplein 50, 3015 GE Rotterdam, The Netherlands

## Introduction

The organization of chromosomes into inaccessible and accessible regions is hypothesized to underlie the ability of the genome to function robustly and accurately across a variety of cell types and conditions. Recent developments in sequencing-based chromosomal contact mapping (Hi-C, Lieberman-Aiden *et al*, 2009; 5C, Dostie *et al*, 2006; 4C-seq, van de Werken *et al*, 2012) have greatly refined previous models of chromosomal organization, identifying topological domains that encompass multiple genes (averaging 1 Mb in mouse, Dixon *et al*, 2012; Nora *et al*, 2012; and 100 Kb in *Drosophila*, Sexton *et al*, 2012) and correlate with distinct gene activity profiles (Andrey *et al*, 2013) and epigenetic characteristics. It has been suggested that these domains are fundamental building blocks that support modular and compact chromosomal architectures. However, many questions regarding their functional roles and the mechanisms that define domain borders and drive their intrinsic structure remain unanswered.

The highly conserved cohesin complex has fundamental roles in chromosome biology, which include sister chromatid cohesion and DNA repair (Nasmyth and Haering, 2009). The core complex is a tripartite ring composed of Smc1, Smc3 and Rad21/Scc1 subunits, which encircle and physically tether newly replicated sister chromatids (Gruber *et al*, 2003). Sister chromatid cohesion is maintained until the onset of anaphase, at which point cohesin is fully removed from chromatin and sister chromatids can segregate into daughter nuclei. Many additional accessory proteins have been identified, which are necessary to regulate the loading (Ciosk *et al*, 2000), stabilization (Skibbens *et al*, 1999; Toth *et al*, 1999; Kueng *et al*, 2006) and removal (Hartman *et al*, 2000; Uhlmann *et al*, 2000; Waizenegger *et al*, 2000) of the complex from chromatin. In addition, cohesin proteins bind to chromatin during interphase and have been shown to co-localize with the DNA binding protein CTCF (Parelho *et al*, 2008; Rubio *et al*, 2008; Wendt *et al*, 2008) where they are required to mediate chromatin loops at select candidate sites in the genome (Hadjur *et al*, 2009; Mishiro *et al*, 2009; Nativio *et al*, 2009; Degner *et al*, 2011; Seitan *et al*, 2011).

Although cohesin's ability to facilitate chromosomal loops between CTCF-bound DNA elements has been studied at a number of selected genomic regions, it is currently unclear to what extent cohesin promotes a global network of interactions between any two neighbouring CTCF sites in the genome or only between specific elements for the purposes of gene regulation at individual loci. Analysis of Hi-C and 5C contact maps has suggested that CTCF and cohesin are enriched at borders of topological domains (Dixon *et al*, 2012; Nora *et al*, 2012; Sexton *et al*, 2012; Phillips-Cremins *et al*, 2013) implying a role for cohesin in domain demarcation. Other studies have reported that cohesin–CTCF sites are associated with loops surrounding promoter-enhancer

modules, while CTCF-free cohesin sites have been shown to mediate interactions between enhancers and promoters (Kagey *et al*, 2010; Demare *et al*, 2013; Phillips-Cremins *et al*, 2013).

To determine the global significance of these observations, a comprehensive understanding of the role of cohesin proteins in the establishment and maintenance of chromosomal domains and their internal structures is required. To perform such a comprehensive analysis, appropriate quantitative methodologies must be used in order to build a high-resolution framework that will allow one to distinguish between high specificity cohesin-dependent regulatory contacts and the possible global architectural role of the complex concurrently.

We have analysed chromosome architecture systematically and on a genome-wide basis in wild-type and cohesin-deficient neural stem cells (NSCs) using a combination of Hi-C, high-resolution 4C-seq and 3D DNA FISH. Quantification of chromosomal contacts at multiple scales showed that cohesin/CTCF co-occupied sites are focal points of chromosomal contact insulation, associated with both the borders of topological domains and finer-scale structures within such domains. Our analysis suggests that domain demarcation arises from a remarkably selective and complex hierarchy of cohesin/CTCF-anchored long-range interactions. Importantly, cohesin-deficient post-mitotic nuclei exhibit global architectural changes associated with a decrease in long-range cohesin/CTCF contacts, universal relaxation of domain structure and nuclear decompaction. These structural changes are accompanied by extensive perturbation of gene expression involving not only a deregulation of cohesin-bound genes, but also a widespread transcriptional response of cohesin-free genes, likely as a result of domain relaxation. Taken together, these observations show that selective cohesin/CTCF contacts constitute a key mechanism underlying chromosomal domain architecture, and suggest that this architecture functions to stabilize mammalian transcriptional programmes.

## Results

### Hi-C analysis of proliferating NSCs and their post-mitotic progeny

In order to study chromosomal organization and the contribution of cohesin proteins to domain structure in interphase chromosomes, we generated clonal populations of proliferating NSCs (Conti *et al*, 2005) from mouse embryonic stem cells (ESCs) and then further differentiated NSCs into populations of post-mitotic astrocytes (ASTs) by exposure to BMP4. After validation of key differentiation markers and cell-cycle distribution in the AST and NSC cultures (Supplementary Figure S1), we prepared genome-wide chromosome conformation capture (Hi-C) libraries, sequencing 85–130 million tag pairs per library in biological replicates, followed by filtering and normalization of Hi-C ligation products to remove biases (Supplementary Figures S2 and S3). In agreement with previously published Hi-C studies (Dixon *et al*, 2012; Nora *et al*, 2012; Sexton *et al*, 2012), both the AST and NSC maps (as well as a Hi-C map generated from $G_1$-purified NSC cells; Supplementary Figure S4) exhibited the characteristic decrease in contact probability with increased genomic separation (Supplementary Figure S5) and recapitulated the topological domain structures that have been recently described (Figure 1A). Given the current Hi-C

sensitivity, the AST and NSC Hi-C maps were found to be highly correlated (Supplementary Figure S6), allowing us to perform subsequent analysis of chromosome structure in the two systems in parallel. As previously observed, topological domains cluster into transcriptionally active or passive classes (Lieberman-Aiden *et al*, 2009; Sexton *et al*, 2012) (Figure 1A; Supplementary Figure S7). However, closer examination of the contact profiles within domains reveals that these two classes are strikingly different in their internal structure. Passive domains are typically large with homogeneous internal contact profiles, whereas active domains are smaller with complex internal contact profiles (Figure 1A, zoomed panels). The average contact intensity between pairs of elements within active domains is two-fold higher compared to passive domains, with a higher variance (Figure 1B). Thus, the AST and NSC Hi-C contact maps offer an opportunity to explore both the mechanisms that underlie domain demarcation and those that facilitate internal domain structure.

### Cohesin density is correlated with structural complexity within active Hi-C domains

We mapped Rad21 binding sites using ChIP-seq in ASTs and NSCs and compared them to published data sets of mouse CTCF binding sites (Shen *et al*, 2012). In agreement with previous observations (Parelho *et al*, 2008; Rubio *et al*, 2008; Wendt *et al*, 2008), we found that the majority of Rad21 binding sites are enriched for CTCF (Supplementary Figure S8). Systematic comparison of co-occupied cohesin/CTCF binding sites with Hi-C maps revealed an association between factor binding and the complexity of the domain structure. First, we observed that the density of binding sites is significantly higher in active compared to passive domains (Figure 1C). Second, we observed an enrichment of binding sites at previously described domain borders (Supplementary Figures S9 and S10). Finally, we detected an inverse correlation between the number of binding sites separating two chromosomal elements and the likelihood that those elements will interact (Figure 1D). These observations indicated that cohesin/CTCF binding sites could act as contact insulators that prevent chromosomal interactions across them, not only at domain borders but also within domains.

### Contact insulation around cohesin/CTCF co-occupied sites is observed at multiple scales

To further characterize chromosomal contacts around cohesin/CTCF binding sites, we next studied the average contact distributions around these sites using high-resolution quantitative analysis of Hi-C maps. We pooled together contact frequency data from individual restriction fragment pairs around thousands of cohesin/CTCF sites and thus were able to enhance the resolution of the Hi-C map. We used a computational approach that allowed for sensitive quantification of chromosomal *contact insulation* (measured by the decrease in contact probability) between multiple elements separated by a cohesin/CTCF site. We performed the analysis at multiple distance scales (Figure 1E; Supplementary Figure S11) to describe both megabase-sized domains that have been previously identified (see 640 Kb band) and extensive contact insulation at finer scales (see 80 Kb band).

We observed robust insulation around cohesin/CTCF sites at all distance scales, reflected by a peak-to-trough ratio of over 1.5-fold (Figure 1F). This analysis also indicated that

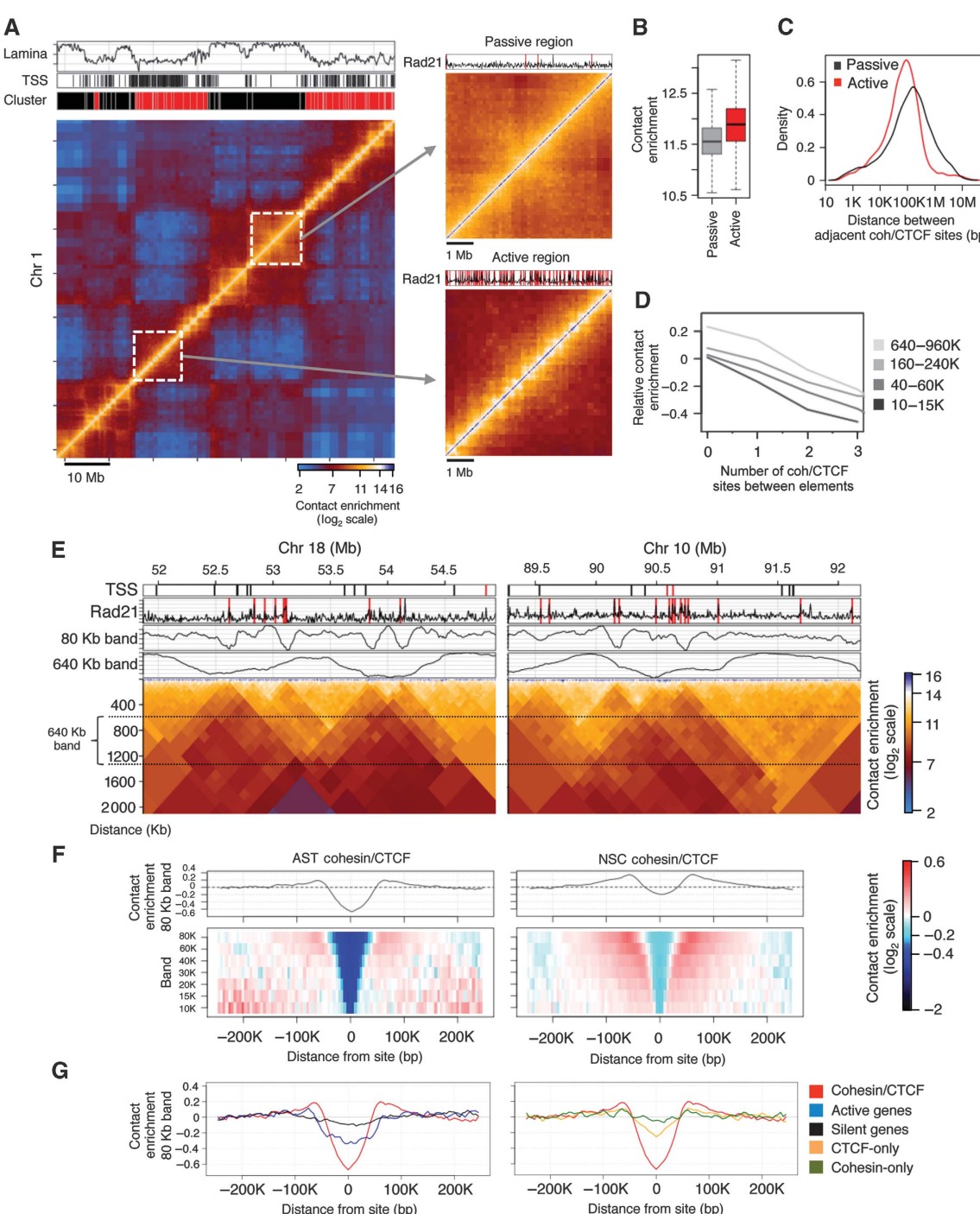

**Figure 1** Cohesin/CTCF sites anchor chromatin loops of multiple sizes. (**A**) Hi-C contact map of a 70-Mb region on chromosome 1 in NSCs, coloured according to technically corrected contact enrichment (Yaffe and Tanay, 2011). Active chromatin (red cluster) is enriched for transcription start sites (TSSs) and depleted for lamina interactions, whereas passive chromatin (black cluster) exhibits the opposite characteristics (see also Supplementary Figure S7). Insets: zoomed-in active and passive regions. (**B**) Genome-wide distributions of contact enrichment within active and passive domains, for genomic distances between 60 and 180 Kb ($\log_2$ scale). (**C**) Distribution of the distance between adjacent cohesin binding sites within active and passive domains (Kolmogorov–Smirnov D $=0.1924$, $P \ll 10^{-10}$). (**D**) Pairs of elements were grouped according to the number of binding sites separating them (*x* axis) and their genomic distance. The average contact enrichment is shown, relative to the group genomic average (*y* axis, $\log_2$ scale). (**E**) Hi-C contact maps of a 3-Mb region on chromosomes 18 and 10 in NSCs (the matrix has been rotated by 45 degrees). Rad21 binding sites and TSSs are shown. Average contact enrichments represent the intensity of interactions crossing each genomic locus, while controlling for genomic distance (tracks for the 80 or 640 Kb bands are shown). (**F**) Multi-scale colour-coded heatmaps of the average contact intensity around cohesin/CTCF binding sites, using a series of high-resolution bands ranging from 10 to 80 Kb in both AST and NSCs. (**G**) Left panel, comparison of the average contact intensity for Rad21/CTCF sites (red) with that for active genes (blue) and silent genes (black) at the 80-Kb band and right panel, with that for CTCF-only (yellow) or Rad21-only (green) sites.

cohesin/CTCF sites have increased contact intensities with elements immediately flanking them. For comparison, analysis of contacts around active transcription start sites (aTSSs)

indicated lower levels of insulation at the 80-Kb scale (Figure 1G, left panel). To test whether this was common to all cohesin/CTCF sites or only a subset, we grouped cohesin

sites according to their extent of insulation (based on the 80-Kb band) and repeated the insulation analysis for the top, middle two and bottom quartiles (Supplementary Figure S12). We observed consistent insulation signatures at the 10–15 Kb bands, suggesting that even if cohesin sites do not engage in insulation at larger scales (i.e., borders of topological domains), they are still able to influence their local contact environment. Surprisingly, we detected minimal insulation signatures at all distance scales for CTCF sites lacking Rad21 as well as Rad21 sites lacking CTCF (Figure 1G, right panel; Supplementary Figure S13). These results confirm that cohesin/CTCF co-occupied sites engage in contact insulation throughout the genome at multiple scales.

### Loss of cohesin leads to global perturbation of chromosomal insulation

Next, we wanted to test whether cohesin proteins are functionally required for the contact insulation observed at their binding sites. $Rad21^{WT/WT}$ and $Rad21^{Lox/Lox}$ NSCs were induced to differentiate into ASTs for 24 h at which point >90% of the population had exited the cell cycle (Figure 2A; Supplementary Figure S14), enabling us to eliminate the confounding effects of cohesin's role during cell division. We treated $Rad21^{Lox/Lox}$ ASTs with Adenovirus-CMV-Cre (Adv-Cre) to induce a deletion within the Rad21 gene and within 96 h, Rad21 protein levels had dropped to 11% of control levels (Figure 2B). Rad21-deficient ASTs remained synchronized in $G_1$ and there was no excess cell death associated with loss of the protein at the time points analyzed (Supplementary Figure S14). Hi-C contact maps generated from $Rad21^{Lox/Lox}$ and $Rad21^{\Delta/\Delta}$ ASTs revealed global differences in chromosomal contacts both within and between domains (Figure 2C; Supplementary Figures S15–S17). A representative differential contact map, colour-coded according to the difference in contact intensity between $Rad21^{\Delta/\Delta}$ and $Rad21^{Lox/Lox}$ ASTs, exemplifies that cohesin depletion is characterized by decreased intra-domain contacts (blue) and increased inter-domain contacts (red), while domain borders remain similar (Figure 2D). These observations represent a global trend, demonstrated by the sharper decrease in contact probability as a function of distance in the knockout (Supplementary Figure S18). Furthermore, we found no evidence for cohesin depletion to have a differential effect in active compared to passive regions (Supplementary Figure S18). Our results suggest that in the absence of functional cohesin, chromosome structure is globally perturbed, irrespective of activity state.

Contact insulation analysis in $Rad21^{Lox/Lox}$ and $Rad21^{\Delta/\Delta}$ ASTs shows that in the absence of cohesin, there is a significant reduction in insulation at cohesin/CTCF binding sites, accompanied by a loss of contacts between those sites and their surroundings (Figure 2E, left side). This effect is observed at multiple scales and irrespective of the basal extent of insulation (Supplementary Figure S19), while insulation around aTSSs does not change (Figure 2E, right side). Importantly, Hi-C maps generated from Rad21-deficient NSCs showed similar trends in comparison to their controls (Supplementary Figure S20), and analysis of $Rad21^{WT/WT}$ ASTs expressing Adv-Cre showed minimal disruption of domain structure compared to untreated $Rad21^{WT/WT}$ ASTs (Supplementary Figure S21). Thus, Hi-C analysis of Rad21-

deficient cells supports a role for cohesin in maintaining the proper organization of interphase chromosomes.

### Domain decompaction in cohesin-deficient cells

To confirm that the Hi-C data generated from Rad21-deficient cells indeed reflect domain decompaction, we applied 3D DNA FISH to a 6-Mb region of chromosome 1 shown in Figure 2C, in which the AST Hi-C map is altered upon Rad21 loss. According to the Hi-C data, intra-domain contacts in the region decrease, whereas inter-domain contacts are increased in $Rad21^{\Delta/\Delta}$ cells. 3D DNA FISH analysis of two probes separated by 500 Kb and designed to hybridize to the borders of the same active domain indicated a significant increase in inter-probe distance in Rad21-deficient nuclei (probes C–D, Figure 2D and F). The same trend was observed for another pair of probes separated by 1 Mb (Pair B–D). Conversely, we detected a significant decrease in inter-probe distance between probes separated by 4 Mb and positioned within two separate nearby domains (probes A–E). These results are consistent with our Hi-C data and confirm that the loss of cohesin results in decreased intra-domain contacts and increased interactions between neighbouring domains. The increase in inter-domain contacts observed here could result indirectly from interactions between neighbouring decompacted domains as opposed to direct interactions (Supplementary Figure S22). Importantly, the same observations were confirmed with Rad21-deficient NSCs (Supplementary Figure S23).

To assess decompaction in an independent way, we used a series of neighbouring BAC probes to paint a 1.9-Mb domain and estimated the volume of the hybridization signal (representative of the three-dimensional domain). We found that $Rad21^{\Delta/\Delta}$ ASTs have significantly larger domain volumes compared to controls (29% increase) (Supplementary Figure S24), further supporting our observations from the inter-probe distance analysis. Finally, we also observed that nuclear volumes of Rad21-deficient ASTs were 26% larger on average compared to controls (Figure 2G), suggesting that domain decompaction resulting from cohesin loss affects nuclear structure on a global scale. Altogether, the FISH analysis validates the results obtained by Hi-C and confirms that in the absence of functional cohesin, chromatin domains throughout the genome become decompacted and more prone to inter-domain interactions.

### A selective cohesin interaction network underlies a hierarchy of topological domains

To further characterize the mechanisms by which cohesin proteins facilitate domain demarcation, we aimed to map cohesin-mediated contacts at higher resolution than currently available from Hi-C and 5C maps. Using the Hi-C maps as a guide, we designed 4C viewpoints to cohesin/CTCF binding sites, which were either proximal to strong domain borders or within a domain. We performed 4C-seq experiments using two rounds of 4 bp cutters (van de Werken *et al*, 2012) to generate high-resolution contact profiles, allowing us to measure contact frequencies from the viewpoint of individual binding sites. As shown in Figure 3A and B, we discovered a remarkably preferential network of cohesin/CTCF contacts that underlies the Hi-C domain structure. For example, the cohesin/CTCF sites on either side of the domain border depicted in Figure 3A are engaged in highly specific

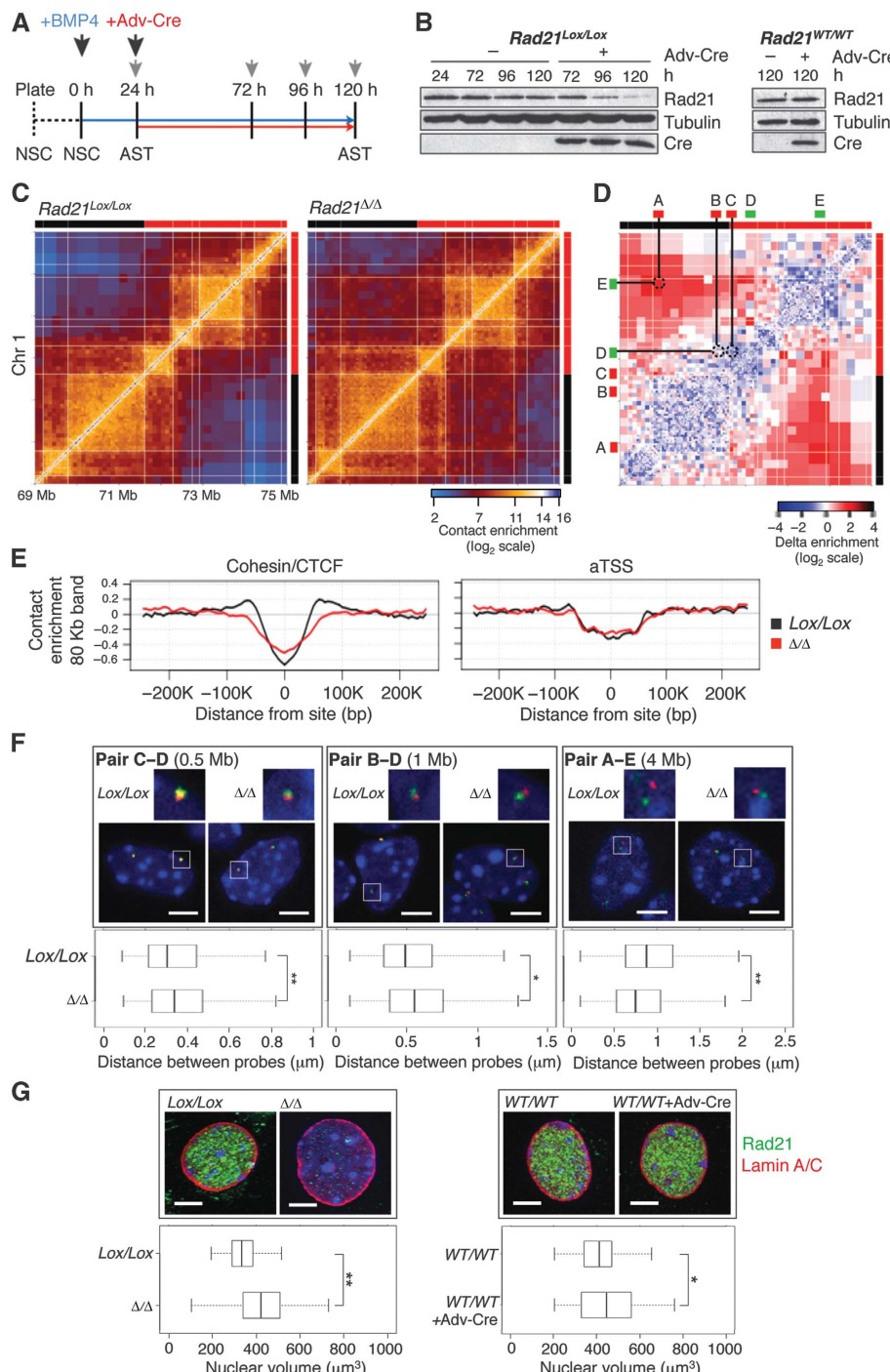

**Figure 2** Loss of a functional cohesin complex perturbs nuclear structure. (**A**) Differentiation scheme of *Rad21*$^{Lox/Lox}$ NSCs to *Rad21*$^{\Delta/\Delta}$ ASTs. NSCs were grown in the presence of EGF/FGF (dotted line) followed by replacement with BMP4 (blue arrow). Twenty-four hours later, ASTs were infected with Adv-Cre (red arrow). Grey arrowheads represent sample collection time points. (**B**) Immunoblot analysis of changes in Rad21 protein levels following Adv-Cre addition to *Rad21*$^{Lox/Lox}$ and *Rad21*$^{WT/WT}$ AST cultures. Eighty-nine percent of the Rad21 protein is lost 96 h after infection specifically in *Rad21*$^{Lox/Lox}$ cells (quantified with ImageJ). (**C**) Hi-C contact maps of a 6-Mb region on chr 1 from *Rad21*$^{Lox/Lox}$ and *Rad21*$^{\Delta/\Delta}$ AST cells. (**D**) A delta contact map colour-coded according to the difference in normalized contact intensity between the two conditions in (**C**). Indicated are the locations of 3D DNA FISH probes (**A–E**) using colour bars, probes have been paired on the matrix using straight lines. (**E**) Average contact intensities (80 Kb band) across cohesin/CTCF binding sites and aTSSs, comparing *Rad21*$^{Lox/Lox}$ ASTs (black) to *Rad21*$^{\Delta/\Delta}$ ASTs (red). (**F**) Representative confocal images of nuclei and quantification of three-dimensional inter-probe distances for the indicated probe pairs in *Rad21*$^{Lox/Lox}$ and *Rad21*$^{\Delta/\Delta}$ ASTs (Volocity software). DNA has been counterstained with DAPI (blue) and probes are labelled with DIG (green) or biotin (red). Genomic distances between probes are indicated. White boxes show the regions that have been zoomed in. Whiskers and boxes indicate all and 50% of values, respectively. Central bold bars represent the median. The asterisk indicates a statistically significant difference as assessed using a Wilcoxon Rank-Sum and a Signed-Rank Test (medians: pair C–D 0.30/0.34 microns, *P* = 0.01; pair B–D 0.49/0.55 microns, *P* = 0.04; pair A–E 0.87/0.75 microns, *P* = 0.009). (**G**) Maximum projections from confocal z-stacks of *Rad21*$^{Lox/Lox}$ and *Rad21*$^{WT/WT}$ AST nuclei treated with or without Adv-Cre and stained for Rad21 (green) and Lamin A/C (red). DNA has been counterstained with DAPI (blue). Quantifications and statistical analysis were done as above. Median nuclear volumes − 334 (*Lox/Lox*) versus 422 ($\Delta/\Delta$), *P* = 1.3 × 10$^{-10}$ and 412 (*WT/WT*) versus 445 (*WT/WT*+Adv-Cre), *P* = 0.04. Each experiment was repeated a minimum of two times (*n* > 170/condition). Scale bar = 5 μm. \**P* > 0.01, \*\**P* < 0.01.

contacts with other cohesin/CTCF sites within their respective domains. Remarkably, these sites do not interact with cohesin/CTCF sites located in the adjacent domains despite their close proximity on the linear chromosome (Supplementary Figure S25). In another example shown in Figure 3B, a cohesin/CTCF binding site at a domain border interacts preferentially with a cohesin/CTCF site at the other edge of the domain, which is 2 Mb away, strikingly skipping over several interacting cohesin binding sites (Supplementary Figure S25). These observations are consistent with the idea that domains are folded in a hierarchical fashion by highly selective cohesin/CTCF interactions. In contrast to these cases, additional 4C viewpoints chosen at sites that are not bound by both cohesin and CTCF did not engage in long-range interactions (Supplementary Figure S26). The functional role of cohesin proteins in anchoring long-range contacts that are necessary for proper domain structure is underlined by 4C-seq analysis of $Rad21^{\Delta/\Delta}$ cells. The results indicate that there is a significant decrease in the intensity of the above described cohesin–cohesin contacts (Figure 3A and B). Intriguingly, we also observed that the decompaction of adjacent domains can be accompanied by an increase in contacts between cohesin/CTCF sites in separate domains (see Supplementary Figure S25, right side, bottom bait). Together, these examples show that cohesin/CTCF sites selectively form long-range loops, which function to demarcate domains and define their complex internal structures.

### Intra-domain cohesin/CTCF contacts are perturbed in cohesin-deficient cells

Next, we returned to the Hi-C maps to assess whether the preferential cohesin/CTCF contacts observed in the 4C-seq examples represent a global trend. We focussed on pairs of cohesin/CTCF binding sites (separated by 100–200 Kb) and computed the mean Hi-C contact intensity between all 2 Kb genomic bins located up to 40 Kb upstream and downstream of each binding site. The data were pooled for thousands of pairs, quantifying average contact intensities at the point of the cohesin/CTCF interaction and the regions flanking it (Figure 3C). Consistent with the 4C-seq analysis, this confirmed that cohesin/CTCF sites preferentially contact one another globally in a cohesin-dependent fashion, provided that both sites are contained within the same domain. We refined this observation by estimating cohesin/CTCF contact intensities for different ranges of genomic separation in active and passive domains. A significant 2-fold cohesin-dependent enrichment was observed for interactions between sites separated by 100 Kb or more within a domain (Figure 3D, black and red curves). Similar analysis of inter-domain cohesin/CTCF contacts suggests that such enrichment can be supported for a limited distance range (<1 Mb for active domains, <2 Mb for passive domains; Figure 3E). In contrast to intra-domain contacts, which are lost following *Rad21* knockout, cohesin depletion leads to a shift in the enrichment distances of inter-domain contacts (Figure 3E, red curves peak enrichments), reminiscent of the domain decompaction analysis discussed above. Again, analysis of NSC data confirmed these observations (Supplementary Figure S27).

Interestingly, no significant contact enrichments were detected between cohesin/CTCF sites and non-cohesin/CTCF sites (Figure 3D and E, dashed lines) or for CTCF-only and cohesin-only sites (Supplementary Figure S28). Furthermore,

analysis of the contact preferences between epigenetic hotspots, including cohesin/CTCF sites, active TSSs, putative enhancer loci, silent TSSs and CTCF-only sites, that are not bound by cohesin suggested that strong intra-domain contacts are unique to cohesin/CTCF binding sites (Figure 3F). Enrichment of long-range inter-domain contacts at distances above 1 Mb was primarily detected between cohesin/CTCF sites in passive domains. In conclusion, Hi-C analysis shows globally that cohesin/CTCF sites anchor long-range contacts within domains and further confirms that cohesin loss disrupts these contacts and the structures associated with them.

### Widespread transcriptional deregulation in cohesin-deficient cells

We used RNA-seq to determine whether the global domain perturbation we observe in cohesin-deficient ASTs has an effect on the transcriptional status of these cells. Genome-wide analysis (Figure 4A) showed remarkably widespread differences in expression between $Rad21^{Lox/Lox}$ and $Rad21^{\Delta/\Delta}$ ASTs with extensive upregulation and downregulation of hundreds of genes. Such extensive transcriptional changes can be indicative of an indirect activation of a general cellular programme (e.g., stress response and differentiation); however, a comprehensive analysis of the genes deregulated as a result of Rad21 deficiency did not reveal a significant overlap with known transcriptional modules nor an enrichment for particular functional categories (Supplementary Table S1), suggesting that ASTs respond to cohesin deficiency in a way that is unlikely to be controlled by common secondary signalling and transcriptional regulators.

Analysis of cohesin/CTCF binding at or near the TSSs of deregulated genes showed that the majority of deregulated genes do not have a cohesin/CTCF binding site within 10 Kb of their TSS (Figure 4B), however there was an enrichment for cohesin/CTCF binding at distances <10 Kb from the TSS (Figure 4C), suggesting that some but not all of the responding genes are direct targets for cohesin/CTCF-mediated gene regulatory loops. We hypothesized that cohesin-dependent perturbation of domain organization leads to widespread gene deregulation of genes not bound by cohesin. This is supported by the fact that genes which are not separated by a cohesin/CTCF site (i.e., positioned in a common loop) are more correlated in their transcriptional response to cohesin loss than genes separated by one or more sites (Figure 4D; Supplementary Figure S29). Thus, the exact positioning of genes within domains may contribute to their transcriptional state, supporting the view that cohesin/CTCF-mediated domain organization functions to stabilize transcriptional programmes.

### Contribution of cohesin/CTCF-based loops to transcriptional stability

In order to place the transcriptional changes described above in the context of perturbed chromosomal contact structures, we generated additional 4C-seq profiles from viewpoints located at cohesin sites positioned at the promoters or within the local environment of upregulated or downregulated genes. The 4C-Seq examples demonstrate that complex intra-domain structures underlie gene expression and that such structures are disturbed in cohesin-deficient cells due to the loss of specific cohesin-anchored interactions. For example, a nested two-loop structure on chromosome 15

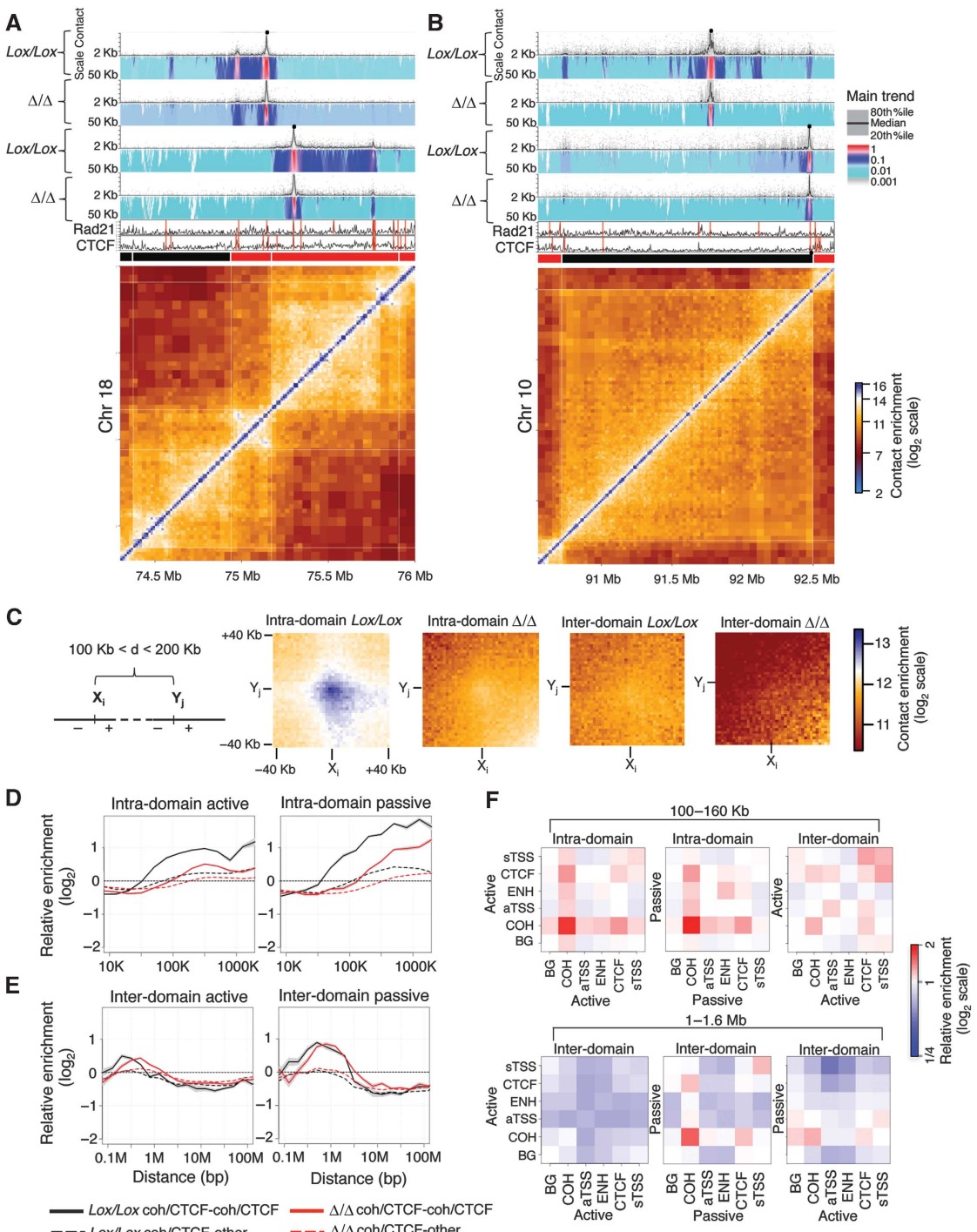

**Figure 3** Cohesins engage in preferential long-range interactions. (**A**) Results from 4C-seq experiments for two viewpoints (black dots) located at cohesin/CTCF binding sites proximal to a border between two active domains, and (**B**) two viewpoints located at a cohesin/CTCF site (right bait) and a cohesin site within a large passive domain or proximal to its border. Each 4C-seq experiment is represented by the median normalized 4C-seq coverage in a sliding window of 5 Kb (top) and a multi-scale domainogram indicating normalized mean coverage in windows ranging between 2 and 50 Kb. Rad21, CTCF binding profiles and the corresponding Hi-C submatrices are also shown (bottom). (**C**) We pooled together Hi-C submatrices, aligned over an interaction between two cohesin binding sites (centre point) and computed the average contact enrichment in high-resolution bins of 2 Kb. Shown are matrices representing interactions between 6058 pairs of cohesin sites that are located in the same domain (intra-domain) and 2771 pairs of cohesin sites located in distinct domains (inter-domain), in both control and Rad21-deficient cells, in all cases controlling for genomic distance (100–200 Kb). Note the specific colour-coding scheme used, designed to highlight the Hi-C dynamic range at these genomic distances. (**D**) Relative intra-domain contact enrichment as a function of distance, when the two sites are <5 Kb away from a cohesin/CTCF binding site (black) or where only one site is <5 Kb away from a cohesin/CTCF site (dashed black). Data for Rad21-deficient cells (red and dashed red) are also shown. (**E**) Same as in (**D**) but showing inter-domain contacts. (**F**) Same as the analysis in (**D**, **E**) but now testing preferential contacts between different classes of genomic loci; cohesin/CTCF sites (COH), cohesin-free CTCF sites (CTCF), active and silent TSSs (aTSS and sTSS, respectively) and putative enhancers based on p300 binding (ENH). Shown are enrichment values computed for intra-domain contacts at genomic distances of 100–160 Kb (leftmost panels), and for inter-domain contacts at genomic distances of 1–1.6 Mb, further classified to interactions between elements within active or passive domains (right).

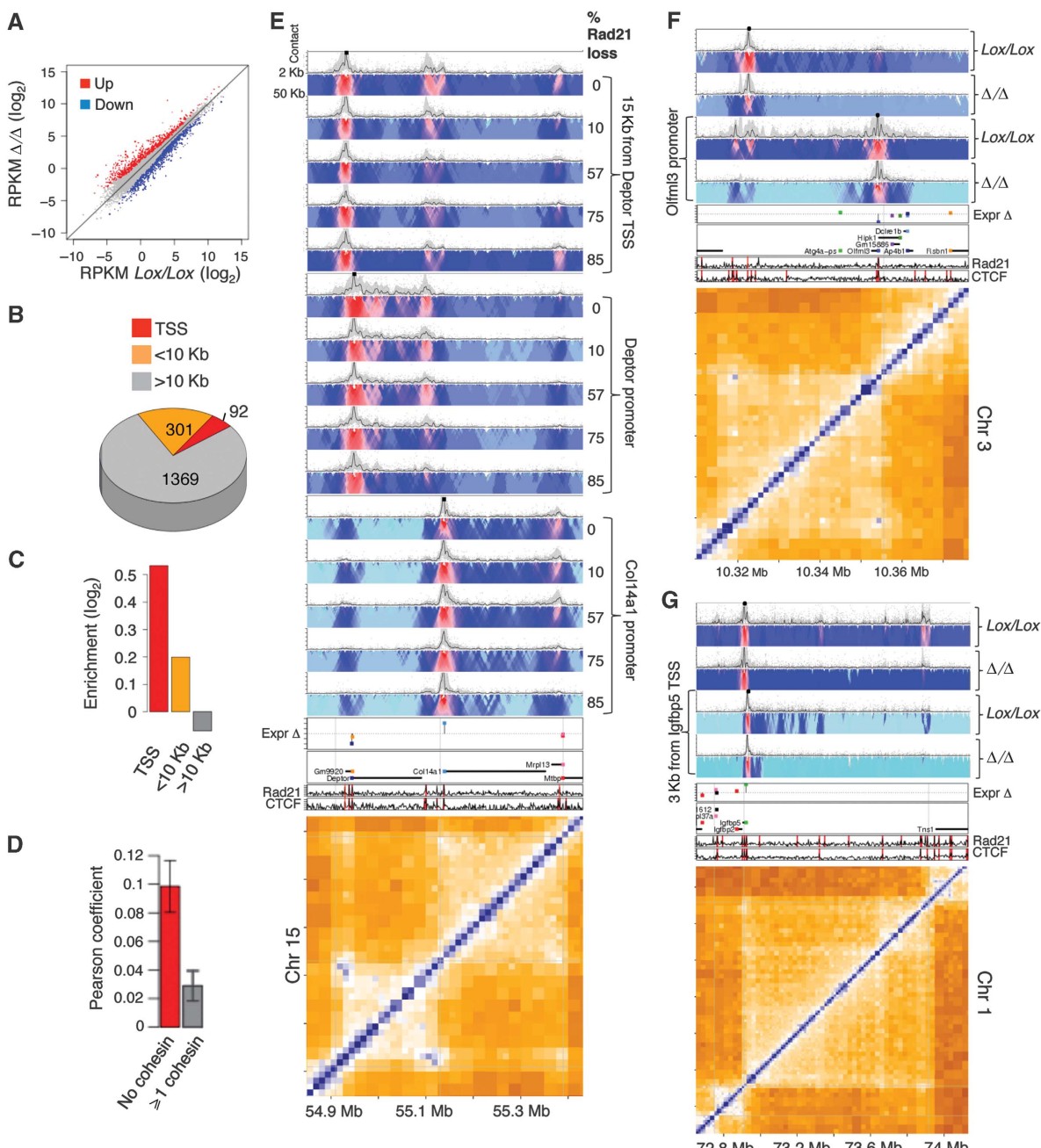

**Figure 4** Large-scale transcriptional deregulation in cohesin-deficient cells. (**A**) Scatter plot comparison between the transcription level of genes in control cells (*x* axis) and cohesin-deficient cells (*y* axis). Genes were classified into 770 upregulated genes (*z*-score > 2, red), 992 downregulated genes (*z*-score < 2, blue) and minimal-change genes (grey). (**B**) Distribution of 1762 deregulated genes according to cohesin/CTCF occupancy at the TSS (< 1 Kb), near the TSS (< 10 Kb) and away from TSS (> 10 Kb). (**C**) Enrichment of the number of deregulated genes in the groups defined in (**B**), over a background composed of all genes. Deregulated genes with cohesin/CTCF at the TSS are enriched by 44%. (**D**) Pearson correlation of the transcriptional response to cohesin knockout of gene pairs, which are 100–200 Kb apart and have no separating cohesin/CTCF site (red). The correlation for pairs of genes which are separated by at least one site is shown as a control (grey). (**E**) 4C-Seq viewpoints positioned (from left to right) at a cohesin/CTCF site 15 Kb upstream of the *Deptor* TSS, 830 bp from the *Deptor* TSS or 1.4 Kb upstream of the *Col14a1* TSS. Shown are the 4C-Seq profiles during a time course of *Rad21* deletion for each viewpoint, which reveal a progressive loss of cohesin–cohesin contacts with decreasing cohesin protein levels. Shown is the % drop in Rad21 protein levels at each time point based on a quantitative western blot analysis. (**F**) 4C-Seq viewpoints positioned 580 bp from the TSS of the downregulated *Olfml3* gene (right bait) as well as at a Rad21/CTCF binding site 300 Kb away. These sites interact according to the Hi-C data and this interaction is specifically lost in Rad21-deficient cells. (**G**) 4C-Seq viewpoints positioned 3.1 Kb from the TSS of the upregulated *Igfbp5* gene (right bait) and a cohesin/CTCF site 10 Kb away from the TSS. The latter preferentially interacts with the cohesin/CTCF site at the other edge of this large domain. This interaction is lost in the mutant. The ChIP-Seq tracks for Rad21, CTCF and TSS locations and change in expression are also shown.

isolates the robustly expressed mTOR inhibitor gene *Deptor* from its neighbouring silent gene *Col14a1* (Figure 4E). A progressive drop in cohesin protein levels over time (Supplementary Figure S30) leads to the progressive relaxa-

tion of this loop structure, downregulation of *Deptor* (2.6-fold) and upregulation of *Col14a1* (7.1-fold). Similarly, loss of cohesin–cohesin loops at the endothelin-converting enzyme-1 (*Ece1*) (Supplementary Figure S30) and olfactome-

din-like protein 3 precursor (*Olfml3*) loci (Figure 4F) is associated with transcriptional repression (2-fold and 4.1-fold, respectively). On the other hand, relaxation of the domains encompassing the insulin-like growth factor-binding protein 5 (*Igfbp5*) and regulator of G-protein signalling 3 (*Rgs3*) loci (Figure 4G; Supplementary Figure S30) is associated with transcriptional upregulation (2.2-fold and 1.6-fold, respectively). Interestingly, there is a slight preference for downregulation of genes that are highly expressed in control ASTs and conversely, for upregulation of lowly expressed genes indicating that cohesin might regulate gene expression noise (Supplementary Figure S29). Altogether, these examples support a role for cohesin/CTCF-mediated long-range interactions in stabilizing transcriptional programmes within well-organized domain structures.

## Discussion

We present several lines of evidence in support of a central role for cohesin in the organization of chromosomal domain structure. Using Hi-C in NSC and AST cells, we show that chromosomal domain architecture is tightly correlated with cohesin/CTCF binding sites, and that in cells lacking functional cohesin complexes, the stability of this architecture is perturbed. Using 3D DNA FISH, we demonstrate that the changes in domain structure of cohesin-deficient cells identified by Hi-C reflect domain decompaction. Using high-resolution 4C-seq, we show that cohesin/CTCF sites interact preferentially to define both intricate loop structures within domains and the borders of megabase-scale chromosomal domains. In Rad21-deficient cells, many of these preferential contacts are lost, accompanied by a general relaxation of the chromosomal domain structure. Thus, domain decompaction comes about as a result of the reduction in cohesin/CTCF distal contacts, which in turn results in more non-specific contacts between domains, indirectly reducing the effective insulation around cohesin binding sites.

The recent discovery of Hi-C contact domains (Dixon *et al*, 2012; Nora *et al*, 2012; Sexton *et al*, 2012) has provided new insights into the organization of genetic information on chromosomes. Our results here provide a mechanistic basis to explain the organization of domains in mammalian chromosomes. CTCF may serve as the initial binding factor, defining a grid of potential insulation sites based on high specificity sequence motifs. Cohesin complexes, and possibly additional components, are then recruited to the CTCF grid, and engage in preferential interactions that give rise to long-range chromosomal loops, which effectively have an insulatory effect and thus organize chromosomes into domains. According to our data, CTCF sites lacking cohesin are neither involved in significant insulation nor do they themselves exhibit long-range interactions and may either serve as dormant insulation hotspots, or function in other cell-type specific contexts. Importantly, the source of specificity of interactions between cohesin/CTCF sites is still unresolved, as it may be either driven indirectly by various regulated processes, such as transcription and replication, or regulated directly via additional uncharacterized mechanisms.

A study published while this work was under review (Seitan *et al*, 2013) suggested that cohesin-deficient thymocytes do not exhibit global architectural changes leading to the conclusion that cohesin is functioning downstream of domain architecture and is not causal for its formation. According to our data, Rad21 knockout results in a global change in chromosomal domain architecture, which is reflected by Hi-C, 4C and 3D-FISH and correlates with gene expression changes. We suggest that the apparent contradiction between the thymocyte and astrocyte study conclusions can be readily explained by the different analysis methodology used. Seitan and colleagues analyse contact frequencies in large genomic bins of 100–140 Kb, a resolution that makes it difficult to observe many of the effects we describe. Our analysis is based on normalizing and pooling genomic landmarks to generate sensitive and quantitative reconstruction of the contact structure changes around cohesin binding sites. This method combined with high-resolution 4C-Seq analysis of individual cohesin binding sites has allowed us to observe the complexity associated with cohesin-dependent chromosome organization.

Following Rad21 knockout, post-mitotic AST nuclei show a global relaxation, but not an abolished domain structure. While this phenotype was essential for the quantitative characterization of the functional role of cohesin/CTCF contacts, it also raised the question of which additional mechanism contributes to the maintenance of domain borders following loss of the majority of cohesin protein. We hypothesize that the residual cohesin complexes on chromatin contribute to the partial preservation of the domain structure, and the gradual degradation of cohesin contact networks we reveal in the 4C-seq time series data supports this idea. It is also possible that other proteins or variants of the cohesin complex help to maintain domain structure. Moreover, it is also likely that domains can be passively maintained, at least for some time, in post-mitotic chromosomes, based only on the prior compaction and organization that was established during exit from the last cell cycle. According to this view, the kinetics of domain relaxation in the absence of cohesin may be affected by numerous factors.

The genome-wide function of cohesin complexes in the organization of chromosomal architecture described here suggests that the effect of cohesin loss on gene regulation may be profound, albeit indirect. If appropriate gene expression depends on the existence of well-organized chromosomal loops and domains to cluster genes and their regulatory elements together, then it can be expected that the global deterioration in chromosomal structure following cohesin loss that we observe would affect many (or even most) genes at some level. Nevertheless, a global effect of chromosomal structures on gene regulation may be mostly associated with maintenance of epigenetic stability and regulation of gene expression noise, rather than classical changes in gene regulation. Moreover, the regulatory effect of cohesin-mediated domain structure may become critical in cycling cells, which must tolerate highly dynamic chromosomal processes and then re-model their chromosome architectures in order to stabilize appropriate gene expression programmes. Further insights from Hi-C and 4C-seq studies combined with extensive epigenetic data and thorough models of gene regulation will be needed in order to eventually develop a comprehensive and quantitative understanding of the complex ways in which chromosomal architecture sets the stage for gene regulation.

## Materials and methods

Mouse NSCs were generated from ESCs and cloned using a protocol described in Conti *et al* (2005). NSCs were differentiated into post-mitotic ASTs in the presence of BMP4. To delete Rad21, Cre recombinase was expressed in *Rad21*$^{Lox/Lox}$ AST cells. Hi-C and 4C-seq libraries were prepared according to previously described protocols with minor adjustments (Lieberman-Aiden *et al*, 2009; van de Werken *et al*, 2012) and sequenced on Illumina GAII or MiSeq platforms. 3D DNA FISH was done following a published protocol (Eskeland *et al*, 2010) and distance and volume measurements were done using the Volocity (Perkin Elmer) or Imaris (Bitplane) softwares. Computational analysis was based on the probabilistic approach described in Yaffe and Tanay (2011), Sexton *et al* (2012) and van de Werken *et al* (2012) with several extensions. Techniques for enhancing the resolution of Hi-C maps using sensitive pooling and normalization of individual fragment end pairs enabled the construction of insulation diagrams and cohesin–cohesin interaction matrices at a 2 Kb resolution. Detailed experimental protocols are provided in Supplementary Methods.

### Accession code

The data analysed in this study has been deposited in the GEO database with ID number GSE49018.

### Supplementary data

Supplementary data are available at *The EMBO Journal* Online (http://www.embojournal.org).

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

## Acknowledgements

We wish to thank Zoe Webster for ESC preps; Andrew Oldfield for cell-culture support; Lu Zhang, Shujun Luo and Robin Li for help with library preparations and sequencing; Sreenivasulu Kurukuti and David Vetrie (Glasgow University) for help with Hi-C; Tomas Adejumo for cell sorting; Lauren McLaughlin and Anne Vaahtokari for help with microscopy; Ben Martynoga and Francois Guillemot for providing p300 ChIP data in NSCs; Gilad Landan for 4C-Seq advice; and Ephraim Kenigsberg for advice on analysis and data manipulation. We would also like to acknowledge members of the Hadjur and Tanay groups for discussions. This work was supported by the Medical Research Council (G0900491/1 and G1001649) granted to SH, the EPIGENESYS EU NoE (for SH and AT) and the ISF (grant number 1796/12 for AT). AT is an incumbent of the Rich Family CDC.

*Author contributions*: SS and SH initiated the project. SH and SMP created the NSC populations. SS performed the Hi-C, 4C-Seq and 3D DNA FISH experiments, including library preparations. DG performed the RNA-Seq and nuclear microscopy experiments. GPS sequenced all libraries. SS and DG sequenced the 4C-Seq libraries. EY, W-CC and AT processed and statistically analysed the data. EY and AT developed the contact insulation method. SS, EY, AT and SH wrote the manuscript, with contributions from all authors.

## Conflict of interest

The authors declare that they have no conflict of interest.

Seitan V, Faure A, Zhan Y, McCord R, Lajoie B, Ing-Simmons E, Lenhard B, Giorgetti L, Heard E, Fisher A, Flicek P, Dekker J, Merkenschlager M (2013) Cohesin-based chromatin interactions enable regulated gene expression within pre-existing architectural compartments. *Genome Res* (advance online publication, 3 September 2013; doi:10.1101/gr.161620.113)

Seitan VC, Hao B, Tachibana-Konwalski K, Lavagnolli T, Mira-Bontenbal H, Brown KE, Teng G, Carroll T, Terry A, Horan K, Marks H, Adams DJ, Schatz DG, Aragon L, Fisher AG, Krangel MS, Nasmyth K, Merkenschlager M (2011) A role for cohesin in T-cell-receptor rearrangement and thymocyte differentiation. *Nature* **476:** 467–471

Sexton T, Yaffe E, Kenigsberg E, Bantignies F, Leblanc B, Hoichman M, Parrinello H, Tanay A, Cavalli G (2012) Three-dimensional folding and functional organization principles of the Drosophila genome. *Cell* **148:** 458–472

Shen Y, Yue F, McCleary DF, Ye Z, Edsall L, Kuan S, Wagner U, Dixon J, Lee L, Lobanenkov VV, Ren B (2012) A map of the cis-regulatory sequences in the mouse genome. *Nature* **488:** 116–120

Skibbens RV, Corson LB, Koshland D, Hieter P (1999) Ctf7p is essential for sister chromatid cohesion and links mitotic chromosome structure to the DNA replication machinery. *Genes Dev* **13:** 307–319

Toth A, Ciosk R, Uhlmann F, Galova M, Schleiffer A, Nasmyth K (1999) Yeast cohesin complex requires a conserved protein, Eco1p(Ctf7), to establish cohesion between sister chromatids during DNA replication. *Genes Dev* **13:** 320–333

Uhlmann F, Wernic D, Poupart MA, Koonin EV, Nasmyth K (2000) Cleavage of cohesin by the CD clan protease separin triggers anaphase in yeast. *Cell* **103:** 375–386

van de Werken HJG, Landan G, Holwerda SJB, Hoichman M, Klous P, Chachik R, Splinter E, Valdes-Quezada C, Oz Y, Bouwman BAM, MJAM Verstegen, De Wit E, Tanay A, De Laat W (2012) Robust 4C-seq data analysis to screen for regulatory DNA interactions. *Nat Methods* **9:** 969–972

Waizenegger IC, Hauf S, Meinke A, Peters JM (2000) Two distinct pathways remove mammalian cohesin from chromosome arms in prophase and from centromeres in anaphase. *Cell* **103:** 399–410

Wendt KS, Yoshida K, Itoh T, Bando M, Koch B, Schirghuber E, Tsutsumi S, Nagae G, Ishihara K, Mishiro T, Yahata K, Imamoto F, Aburatani H, Nakao M, Imamoto N, Maeshima K, Shirahige K, Peters J-M (2008) Cohesin mediates transcriptional insulation by CCCTC-binding factor. *Nature* **451:** 796–801

Yaffe E, Tanay A (2011) Probabilistic modeling of Hi-C contact maps eliminates systematic biases to characterize global chromosomal architecture. *Nat Genet* **43:** 1059–1065

