## [Review Process File · The EMBO Journal]

Manuscript EMBO-2013-87054

Cohesin-mediated interactions organize chromosomal domain architecture

Sevil Sofueva, Eitan Yaffe, Wen-Ching Chan, Dimitra Georgopoulou, Matteo Vietri Rudan, Hegias Mira-Bontenbal, Steven M. Pollard, Gary P. Schroth, Amos Tanay, Suzana Hadjur

Corresponding author: Suzana Hadjur, University College London

Review timeline:	Submission date:	02 October 2013
	Editorial Decision:	08 October 2013
	Accepted:	08 October 2013

Transaction Report:

Editor: Thomas Schwarz-Romond

Transfer Note	09 October 2013
---------------	-----------------

PLEASE NOTE that this manuscript was transferred from a different journal and the arbitrating referee assessing suitability for The EMBO Journal had access to both the original anonymous comments as well as the point by point response provided by the authors.

Editorial Staff
The EMBO Journal

1st Editorial Decision	08 October 2013
-----------------

Thank you for submitting your manuscript to the EMBO Journal. I involved one new referee who had access to the previous reports and author responses. As you can see below, the referee finds the analysis exciting and supports publication in The EMBO Journal. We are therefore pleased to inform you that your manuscript has been accepted for publication in the EMBO Journal.

REFeree REPORT:

Sofueva and colleagues present a detailed chromosome conformation analysis to uncover the role of cohesin in chromosome topology and gene expression. The study was initially submitted elsewhere and reviewed (with their comments being made available to me). I believe the authors sufficiently addressed most comments of these referees (with the exception of the request of two referees to

provide insight in the dynamic contact changes that may occur between cell types. Here, the authors mention this was not the scope of their work which focuses on cohesin: I agree).

While under review, Merkschlager and colleagues published a story on the same topic in Genome Research. In their letter to the editor, Tanay and Hadjur argue that they have 'come to dramatically different conclusions about cohesin's role in chromosome organization due to the difference in resolution achieved by our methods.' I am not convinced that the conclusions are 'dramatically different'. However, as the authors correctly point out they do provide superior resolution, and they do support their important conclusions by elegant and convincing high resolution 4C-seq experiments, an asset of this study.

The authors state that the improved resolution enables them to draw new conclusions:

- 1) Cohesin binding sites throughout the genome are contact insulators both at local levels (make gene loops) and at hierarchical levels (megabase-sized domain borders).
- 2) We also show, that this effect is unique to cohesin sites only when co-occupied by CTCF, as CTCF-only sites do not anchor loops.
- 3) Domains throughout the genome are decompacted in the absence of cohesin. We show this using our high-resolution Hi-C data as well as 3D microscopy of single cells (FISH techniques) which perfectly supports the 'C' data.

I agree with the authors that this is novel and important new insight into cohesin's role in nuclear architecture, and that the data provided allow drawing these conclusions. Moreover and as pointed out by the initial referees, this is an elegant study, with well performed experiments and state-of-the-art data analysis. Particularly elegant are the time-course 4Cseq experiments, showing the impact of loss of cohesin on individual chromatin loops.

In my opinion, this is an important study that deserves to be published in a top journal like EMBO J as I am convinced it will appeal to its readers.